# Role of Myeloperoxidase, Oxidative Stress, and Inflammation in Bronchopulmonary Dysplasia

**DOI:** 10.3390/antiox13080889

**Published:** 2024-07-23

**Authors:** Tzong-Jin Wu, Xigang Jing, Michelle Teng, Kirkwood A. Pritchard, Billy W. Day, Stephen Naylor, Ru-Jeng Teng

**Affiliations:** 1Department of Pediatrics, Medical College of Wisconsin, Suite C410, Children Corporate Center, 999N 92nd Street, Milwaukee, WI 53226, USA; twu@mcw.edu (T.-J.W.); xgjing@mcw.edu (X.J.); mteng@mcw.edu (M.T.); 2Children’s Research Institute, Medical College of Wisconsin, 8701 W Watertown Plank Rd., Wauwatosa, WI 53226, USA; kpritch@mcw.edu; 3Department of Surgery, Medical College of Wisconsin, 8701 Watertown Plank Rd., Milwaukee, WI 53226, USA; 4ReNeuroGen LLC, 2160 San Fernando Dr, Elm Grove, WI 53122, USA; billy.day@rngen.com (B.W.D.); snaylor@rngen.com (S.N.)

**Keywords:** bronchopulmonary dysplasia, myeloperoxidase, endoplasmic reticulum stress, cellular senescence, N-acetyl-lysyltyrosylcysteine amide

## Abstract

Bronchopulmonary dysplasia (BPD) is a lung complication of premature births. The leading causes of BPD are oxidative stress (OS) from oxygen treatment, infection or inflammation, and mechanical ventilation. OS activates alveolar myeloid cells with subsequent myeloperoxidase (MPO)-mediated OS. Premature human neonates lack sufficient antioxidative capacity and are susceptible to OS. Unopposed OS elicits inflammation, endoplasmic reticulum (ER) stress, and cellular senescence, culminating in a BPD phenotype. Poor nutrition, patent ductus arteriosus, and infection further aggravate OS. BPD survivors frequently suffer from reactive airway disease, neurodevelopmental deficits, and inadequate exercise performance and are prone to developing early-onset chronic obstructive pulmonary disease. Rats and mice are commonly used to study BPD, as they are born at the saccular stage, comparable to human neonates at 22–36 weeks of gestation. The alveolar stage in rats and mice starts at the postnatal age of 5 days. Because of their well-established antioxidative capacities, a higher oxygen concentration (hyperoxia, HOX) is required to elicit OS lung damage in rats and mice. Neutrophil infiltration and ER stress occur shortly after HOX, while cellular senescence is seen later. Studies have shown that MPO plays a critical role in the process. A novel tripeptide, N-acetyl-lysyltyrosylcysteine amide (KYC), a reversible MPO inhibitor, attenuates BPD effectively. In contrast, the irreversible MPO inhibitor—AZD4831—failed to provide similar efficacy. Interestingly, KYC cannot offer its effectiveness without the existence of MPO. We review the mechanisms by which this anti-MPO agent attenuates BPD.

## 1. Introduction

Bronchopulmonary dysplasia (BPD) is a chronic lung disease primarily impacting premature neonates and is complicated by impaired lung development [1,2]. The incidence of BPD in premature infants less than 28 weeks gestational age is approximately 50–70% and is inversely related to gestational age [3]. It has been estimated that in the United States alone, more than 15,000 new BPD patients are diagnosed each year [4]. The annual total cost of BPD was estimated at 2.5 billion USD in the USA alone in 2005 [4]. A recent systemic review estimated the average cost for each BPD premature infant ranged between 21,392 and 1,094,509 USD during birth hospitalization and 46,948 USD during the first year of life. There were also costs for follow-up, home care, and indirect costs of BPD [5]. These put considerable financial pressure on families, medical facilities, and society. Notably, the annual number of new patients has remained unchanged for a considerable period. The primary causal factors of BPD are premature birth and exposure to elevated concentration levels of oxygen produced by mechanical ventilation or other supplemental methods [6]. The diagnostic criteria for BPD continue to evolve. Currently, they are based on clinical criteria such as the need for oxygen at postnatal 28 days of age and radiographic lung abnormalities [7] or, more often, oxygen requirement beyond the postconceptional age of 36 weeks [8]. Current BPD treatments involve a multi-pronged approach to minimize lung injury, promote lung growth, and address complications associated with the condition. Common treatments include supplemental oxygen, surfactants, diuretics, bronchodilators, and steroids [9]. BPD is a complex, multifactorial, and poorly understood disease with limited diagnostic criteria and no specific, safe, and efficacious therapeutic drugs.

The events leading to the onset of BPD are exceedingly complicated. Premature neonates born before 36 weeks of gestation usually do not generate enough endogenous surfactant to keep their lungs open, leading to respiratory distress syndrome (RDS) after birth [10]. Supplemental oxygen, positive pressure respiratory support, and exogenous surfactants are usually given to assist tissue oxygenation. Premature lungs with immature antioxidative systems face an abrupt oxidative stress (OS) challenge beyond their coping capacity at birth. Because of the surfactant deficiency, higher oxygen concentrations are often required to maintain tissue oxygenation, which results in even more OS. Chorioamnionitis [11] and mechanical ventilation [12] further aggravate the OS. OS activates alveolar macrophages to recruit neutrophils from the circulation to remove damaged cells as a form of sterile inflammation [13]. Inflammatory cytokines and chemokines released from the inflammatory cells damage lung cells, inhibit angiogenesis and alveologenesis, and encourage tissue fibrosis, resulting in BPD. Many other contributors to BPD in premature human neonates, such as hemodynamically significant patent ductus arteriosus, necrotizing enterocolitis, and nosocomial infection [14] et al., make studying the BPD mechanism in human subjects extremely difficult.

Although most BPD survivors can come off oxygen treatment before twelve months of age, some of them may stay on oxygen for much longer. Roughly 25–40% of BPD infants will develop pulmonary hypertension [15,16], which is associated with a mortality rate of 48% [17]. BPD survivors may continuously suffer from respiratory problems that require hospital admission in the first two years of life. Poor exercise performance, reactive airway disease, high blood pressure, neurodevelopmental deficits, and poor somatic growth are more commonly seen in BPD survivors [18]. Recent evidence also demonstrated that BPD survivors are prone to developing early-onset chronic obstructive pulmonary disease (COPD) [19]. The association between BPD and early-onset COPD indicates an impaired lung growth trajectory.

Research over the past five decades has resulted in some promising treatments that reduce BPD in premature neonates. Preventing a premature birth is undoubtedly the ideal way, which, unfortunately, is not achievable because of lifestyle choices (e.g., smoking, substance or alcohol use, being underweight or overweight, teenage pregnancy, and advanced maternal age), the prevalence of assisted reproductive technologies, exposure to environmental hazards (e.g., air pollution, toxic chemicals, and secondhand smoking), stress from work and society, and social and medical inequities. Antenatal corticosteroids given to mothers at risk of preterm delivery can reduce the risk of RDS and other prematurity-associated morbidities, but the effect on decreasing BPD remains unclear. Postnatal systemic corticosteroids were once commonly used by clinicians who believed the incidence of BPD could be reduced. Unfortunately, the strong association between its use and severe neurologic deficits [20] led the American Academy of Pediatrics to caution against its routine use [21]. Intramuscular high-dose vitamin A injection has been shown to reduce BPD [22], but the method of application has led to limited acceptance by most practices. However, the attempt to use enteral vitamin A administration failed to show any decrease in moderate to severe BPD [23]. The most promising strategy that has received wide acceptance is early caffeine treatment, which demonstrated a significant reduction in BPD [24] and yielded better long-term neurological outcomes. However, we must remember that the reduction in BPD was found in the secondary analysis of that study. Stem cell treatment has been considered the most promising treatment for BPD. Still, concerns regarding possible vascular occlusion, potential tumorigenicity, and difficulty obtaining the appropriate number of cells and maintaining their quality are waiting to be resolved [25]. Presently, there is no report about the outcome of stem cell therapy in BPD. Early administration of surfactant, permissive hypercarbia or gentle ventilation, early application of continuous positive airway pressure, and aggressive nutritional intervention are other clinical measures that have some evidence to decrease BPD development.

The problem we are facing is the unwavering BPD prevalence after implementing all the above treatments. This may partly be explained by the improved survival of those extremely premature neonates who are at much higher risk of BPD. With the unchanged prevalence of BPD, we need to seek new therapeutic strategies to achieve better BPD management. As human premature neonates face complex confounding variables, animal models become the only option to explore potential mechanisms. Rat and mouse pups are born at the canalicular stage of lung development and enter the first wave of alveolar formation at postnatal day 5 (P5) [26,27]. Rodent lungs have a mature antioxidative defense at birth and cope with hyperoxia (HOX)-induced OS much better than their adult counterparts. Oxygen concentrations above 70% are required to cause the BPD phenotype. Using the HOX-induced rat BPD model, we have obtained extensive information about the role of myeloperoxidase (MPO)-mediated OS in the onset and progression of BPD. We also observed the involvement of endoplasmic reticulum and cellular senescence in BPD.

In this review, we begin by discussing how the BPD definition has evolved over the last six decades, followed by pathological findings in human BPD lungs, risk factors from the antenatal to the postnatal period, biomarkers that might help identify premature neonates at risk of BPD, and mechanisms that contribute to BPD onset and progression. From human and animal studies, we recognize the crucial role of MPO in BPD, which leads to the consideration of MPO inhibitors as new therapeutics for BPD. Using our novel reversible MPO inhibitor—N-acetyl-lysyltyrosylcysteine amide (KYC)—as a research probe, we have identified that a systems pharmacology approach offers a promising new perspective on BPD treatment. Our research has demonstrated the importance of MPO-mediated OS in BPD. We opine that OS is, as we consider it, the most critical contributor to BPD and, hence, a promising therapeutic target.

## 2. Changing Perspectives on BPD

The term “bronchopulmonary dysplasia” was introduced in 1967 by Dr. William Northway, a pediatric radiologist. He observed a pattern of lung conditions in premature infants requiring prolonged respiratory support [28]. Before 1967, the condition was not recognized as a distinct clinical entity. It was described in the context of the respiratory complications and chronic lung diseases associated with premature infants. In particular, those patients who had been treated with mechanical ventilation and oxygen therapy for respiratory distress syndrome (RDS) were sometimes labeled as having “hyaline membrane disease” [28]. See Table 1 for a summary outline of the key changes in perspectives on BPD as defined by landmark publications.

**Table 1 antioxidants-13-00889-t001:** The clinical definition of bronchopulmonary dysplasia (BPD).

Author	Year	Criteria
Northway [28]	1967	Clinical course, X-ray, and histology
Tooley [29]	1979	X-ray and O_2_ use at 30 days, PaO_2_ < 60 mm-Hg in room air or PaCO_2_ < 45 mm-Hg
		28 days of O_2_ exposure with characteristic radiologic findings
Shennan [8]	1988	O_2_ requirement at 36 weeks PMA
Jobe [30]	1999	New BPD versus old BPD
Ehrenkranz [31]	2001	FiO_2_ > 0.21 for ≥28 days
		GA < 32 weeks requires O_2_ at PMA 36 weeks, grade at DOL 56
		Mild: room air
		Moderate: FiO_2_ <30%
		Severe: FiO_2_ > 30% or CPAP or IMV
Walsh [32]	2004	Physiological BPD: O_2_ at precisely 36 weeks’ postmenstrual age
		On PPV or >30% O_2_ with SpO_2_ 90–96%
		Fail to maintain SpO_2_ ≥ 90% for 30 min following the O_2_ reduction test
Higgins [33]	2018	GA < 32 weeks, (+) x-ay findings, and respiratory support at PMA 36 weeks to keep SpO_2_ 90–96%
		Gr 1: CPAP, NIPPV, NC ≥ 3 LPM 21% O_2_; NC 1–2 LPM 22–29% O_2_; NC 1 LPM 22–70% O_2_
		Gr 2: IMV 21% O_2_; CPAP, NIPPV, NC ≥ 3 LPM 22–29% O_2_; NC 1–2 LPM ≥30% O_2_; NC 1 LPM > 70% O_2_
		Gr 3: IMV > 21% O_2_; CPAP, NIPPV, NC > 3 LPM > 30% O_2_
		Gr 3a: Death > 14 days of life < 36 weeks PMA from respiratory failure
Jensen [34]	2019	GA < 32 weeks and grade at 36 weeks PMA or discharge
		Gr 1: NC < 2 LPM
		Gr 2: NC > 2 LPM, CPAP, or NIPPV
		Gr 3: Invasive mechanical ventilation

PMA: postmenstrual age; DOL: day of life; GA: gestational age in weeks; CPAP: continuous positive airway pressure; IMV: intermittent mechanical ventilator; NIPPV: non-invasive intermittent positive pressure ventilator; LPM: liters per minute; NC: nasal cannula.

The original definition of BPD was an oxygen requirement for more than 28 days after birth [29]. This definition might be valid in the era of no exogenous surfactant, and most of the survivable premature neonates were more than 32 weeks of gestation. However, with the majority of extremely premature (22–28 weeks of gestation) neonates surviving hospital discharge today, this definition becomes obsolete. Clinicians slowly transitioned into using oxygen support at the postconceptional (or postmenstrual) age of 36 weeks as the definition for BPD after two decades [8]. This definition of oxygen dependence at the postconceptional age of 36 weeks remains the most widely used criterion for clinical studies. The NICHD Neonatal Network first attempted to standardize the definition in 2001, using a postnatal age of 28 days and a postconceptional age of 36 weeks for premature neonates born at less than 32 weeks of gestation as the criteria, with a grading system included. There are three more versions after 2001. Unfortunately, the diagnosis is heavily influenced by the style of practice, especially how aggressive clinicians are willing to wean off respiratory support. The revised grading system was published in the 2018 NIH workshop report. Researchers felt strongly that the NIH grading system could not predict the long-term outcome, which was what clinicians cared about [35]. Jensen used the extensive NICHD Neonatal Network data to construct a new BPD grading system, gaining popularity among neonatologists (Table 1) [34].

The BPD we face nowadays (new BPD) with improved neonatal management differs from the pre-surfactant era (old BPD). The “old BPD” was seen in larger premature infants after aggressive mechanical ventilation and oxygen needs. Typical findings in pathology included severe large airway injury, interstitial and alveolar edema, extensive small airway disease with alternating areas of overinflation and fibrosis, and pulmonary artery muscularization. The “new BPD” we manage today is typically seen in extremely premature infants with modest ventilation and oxygen use. The pathology findings of new BPD include minimal airway changes with less inflammation and fibrosis, arrested alveolarization, and fewer but abnormal pulmonary vasculatures [36].

BPD is the most common lung complication for premature births, with a prevalence range of 11–50% [1]. Almost 80% of premature neonates born at 22–24 weeks of gestation are diagnosed with BPD [37], and 50% of all neonates born before 28 weeks of gestation are diagnosed with BPD [38], but only 20% of all neonates born after 28 weeks of gestation develop BPD [3]. It was initially believed that surfactant deficiency was the cause of BPD. However, the introduction of exogenous surfactants did not successfully reduce the BPD prevalence. Over the past three decades, we have not seen a meaningful decline in the prevalence of BPD due partly to improved survival rates [39].

Our recent studies have provided new insight into the role of myeloperoxidase-mediated OS in BPD. As this work outlines, a new paradigm is emerging in which oxidative stress and its downstream effects must be considered as a possible paradigm shift in our model and understanding of BPD.

## 3. Pathology of Human BPD

The onset and progression of BPD in premature neonates are intimately connected to the complex and variable development of their lungs [40]. Lung development is divided into embryonic (0–8 weeks), pseudoglandular (9–16 weeks), canalicular (16–24 weeks), saccular (24–36 weeks), and alveolar (36 weeks to 2–8 years) stages [41] (Figure 1). Neonates born before 36 weeks of gestation cannot produce an appropriate amount of surfactant and lack alveoli in the lungs, resulting in RDS. With medical advancement, most extremely premature neonates (<28 weeks) are viable nowadays but require mechanical ventilators and supplemental oxygen to maintain their tissue oxygenation for a minimal number of primitive alveoli. Premature neonates born at 22–24 weeks are considered peri-viable as their lungs contain only respiratory bronchioles, which makes it extremely challenging to provide enough oxygen to the vital organs.

The pathological changes in the pre-surfactant era were characterized by heterogeneity, with severe squamous metaplasia of the airways, marked airway smooth muscle hyperplasia, extensive alveolar septal fibrosis, and thickening of the wall of the pulmonary arteries before exogenous surfactant was available [42]. The leading cause of these changes is mechanical trauma with a lack of surfactant treatment, as almost all BPD infants were supported by mechanical ventilators right after birth. In contrast, in the post-surfactant era with gentle mechanical ventilation and aggressive nutritional support, the BPD phenotype is characterized mainly by less heterogeneity, with mainly alveolar simplification, a reduced and dysmorphic vascular bed with rare epithelial lesions, and mild airway smooth muscle thickening [43]. Inhibited angiogenesis is considered the primary mechanism that causes alveolar simplification. Other than the typical pathologic findings, we can see inflammatory cell infiltration, fibrosis, alveolar wall thickening, hyperinflation, and atelectasis in pathology (Figure 2). Arterial wall thickening, pulmonary vein obstruction, and right ventricular hypertension can be seen if pulmonary arterial hypertension occurs. In rare situations, intrapulmonary arteriovenous shunting can be identified in severe BPD that causes severe hypoxemia [44]. Our recent studies also detected evidence of endoplasmic reticulum (ER) stress and cellular senescence, which we will discuss later.

## 4. Risk Factors for BPD Onset

The complex and variable nature of BPD in individual patients has hindered the identification of risk factors for the disease. An unambiguous understanding of risk factors can help research scientists and clinicians investigate BPD pathophysiology, detect early biomarkers before disease onset, offer effective preventive measures, inform the family about decision-making, and improve outcomes. Unfortunately, it is not uncommon to find contradictory reports in the literature, such as the impact of intrauterine growth restriction, chorioamnionitis, gastroesophageal reflux, et al. We must remember that mortality is a competing variable in analyzing BPD. The composite of BPD and mortality is commonly used as the primary outcome in studies. Several risk factors for BPD have been reported, and we can classify them into three groups, as summarized and listed in Table 2.

### 4.1. Prenatal Risk Factors

a.Intrauterine growth restriction (IUGR)

Although premature neonates often have less severe respiratory distress at birth, they have a higher risk of developing BPD [45]. The most substantial evidence was from the twin study conducted by Groene et al., which showed that smaller twins had higher odds (odds ratio 2.5) than co-twins to have BPD [46]. The mechanism behind this association is believed to be poor growth of the pulmonary vasculature.

b.Lack of antenatal corticosteroids

It seems straightforward that antenatal corticosteroids decrease the incidence of BPD by reducing RDS. Data from the Pediatrix Medical Group showed a decreased incidence of BPD in premature neonates if mothers received antenatal corticosteroids, especially those born at 23- and 24-week gestation [58]. However, the literature had conflicting reports when neurodevelopmental outcomes were also considered [59].

c.Maternal smoking

Maternal smoking during pregnancy is associated with intrauterine growth restriction, an increased risk of intrauterine fetal demise, and altered cardiorespiratory responses. Studies have shown that maternal smoking doubles the risk of BPD in premature neonates born before 34 weeks of gestation. Another study demonstrated that maternal smoking increases the risk of moderate to severe BPD [53]. No doubt, smoking cessation offers the offspring a better outcome.

d.Chorioamnionitis

Chorioamnionitis is considered a significant contributor to preterm birth and is believed to increase the risk of BPD [66,67]. However, some animal and human studies have found that chorioamnionitis may fasten lung maturation or modulate the immune response to decrease the risk of BPD. Some studies have shown that chorioamnionitis protects premature neonates against BPD [68]. The conflicting results make it difficult to determine whether chorioamnionitis increases the risk of BPD.

e.Genetics

Two publications [71,72] first suggested the genetic predisposition to BPD by comparing BPD between monochorionic and dichorionic twins. Higher coherence in monochorionic twins in BPD than in dichorionic twins indicates the contribution of a genetic factor. Other genetic studies also support this assumption [73]. Although this is biologically plausible, later studies could not confirm this association [74].

### 4.2. Risk Factors at Birth

a.Gestational age and birth weight

Gestational age is the most significant predictor of BPD [1,47]. Most cases of BPD occur in premature neonates born before 32 weeks of gestation. As birth weight, in general, correlates with gestational age, it is not a surprise to see body weight as a risk factor for BPD.

b.Gender

Although premature neonates have more males than females, clinicians felt a disproportionately high percentage of male BPD. An animal study showed that female mouse pups respond with more robust epigenomic changes that might protect the lungs against HOX [54]. A recent multi-center retrospective study showed that males account for 59% of all BPD, which seemed to support the existence of sexual dimorphism [55]. However, sex does not appear to be associated with adverse outcomes, such as mechanical ventilation, tracheostomy, or mortality, in that cohort.

c.Level of management

Not all premature neonates will be transferred to or delivered to level 3 or 4 medical facilities. The insufficient workforce, lack of experience, and inadequate support from other subspecialties in lower-level medical facilities can affect the outcomes. One recent study from California revealed that level III NICUs have a similar incidence of BPD as level IV NICUs, while the incidence of BPD was significantly higher in level II NICUs (odds ratio 1.23) [60].

### 4.3. Postnatal Risk Factors

a.Respiratory support

Mechanical ventilator-induced barotrauma and volutrauma have been considered iatrogenic contributors to BPD for decades [48]. Permissive hypercarbia or gentle ventilation strategies have been advocated to reduce mechanical trauma to premature lungs in the hope of decreasing BPD. Some centers advocate the early application of continuous positive airway pressure (CPAP) prophylactically to very preterm neonates after birth. Although one randomized control study did not show the benefit [49], the meta-analysis does show that early or prophylactical CPAP reduces BPD with a relative risk of 0.89 [50]. The oxygen requirement in the first week of life [51] and the peak inspiratory pressure on day 4 [52] were also early predictors for BPD.

b.Infections

Invasive lines are required for managing premature neonates. Although central venous catheters are always placed sterile, there is no guarantee that microorganisms will not enter the circulation. The endotracheal intubation is otherwise a clean but not sterile procedure. These invasive lines increase the chance of nosocomial infections. The incidence of nosocomial infections can range between 6 and 25% or 9–62 infections per 1000 patient days [56]. Inflammatory mediators and OS from nosocomial infections can aggravate damage to the premature lungs. Evidence that supports the contributing role of nosocomial infection in BPD comes from a report showing a reduction in BPD by decreasing nosocomial infections [57]. However, the liberal use of antibiotics is not recommended. The best way to reduce nosocomial infections includes antibiotic stewardship, hand hygiene, and infection control measures.

c.Patent ductus arteriosus (PDA)

Almost all premature neonates are born with a PDA. A PDA contributes to BPD by increasing pulmonary blood flow and causing lung injury and inflammation [61]. Moderate/large PDA is also recognized as a risk factor for BPD-associated pulmonary hypertension [62,63,64]. Interestingly, the prophylactical PDA closure meta-analysis shows no change in BPD [65].

d.Gastroesophageal reflux (GER)

Clinicians have strong feelings about the relationship between GER and BPD [69]. However, GER is commonly seen in premature neonates [70], and no evidence exists that GER increases BPD. The relationship may be a secondary association from prematurity itself.

## 5. Biomarkers for BPD

Early prediction of BPD development would allow the implementation of targeted treatment [75]. Researchers have attempted to identify useful biomarkers that can predict premature neonates at risk of BPD. Potential biomarkers, including gestational age, birth weight, gender, chest radiology, lung ultrasound, and proteomics of tracheal fluids, blood, and urine, have all been reported [76]. Unfortunately, most studies were plagued by small sample sizes or a lack of reproducibility. Some potential biomarkers have had variable and inconsistent changes from study to study. For example, vascular endothelial cell growth factor (VEGF), the signaling protein that stimulates blood vessel formation, had no predictive value in BPD [77]. At this moment, no biomarker can be used reliably to predict which neonates are at risk of BPD or to monitor the progression of BPD.

## 6. Oxidative Stress and General Mechanistic Considerations of BPD Onset and Progression

### 6.1. OS in BPD

Supplemental oxygen, mechanical ventilation, and infections all cause lung OS. There is no doubt that supplemental oxygen and infection increase OS in premature lungs [78]. The mechanism by which mechanical ventilation increases OS in the diaphragm [79] and the lungs [80] remains elusive. Still, the effect is spatially limited and is not affected by the presence of infection. OS activates the alveolar macrophages to recruit circulating neutrophils by releasing chemokines and inflammatory cytokines. After arriving at the alveoli, the infiltrated neutrophils are activated to release their content, including MPO, with subsequent hypochlorous acid (HOCl) generation. HOCl is a highly potent reactive oxidant involved in oxidation and chlorination processes [81]. A characteristic footprint defines MPO-mediated OS due to the formation of 3-chlorotyrosine (Cl-Tyr). Activated macrophages transform into a pro-inflammatory M1 phenotype [82] and upregulate their inducible nitric oxide (^●^NO) synthase (iNOS or NOS2). ^●^NO interacts with superoxide (O_2_^●^^−^) to form the peroxynitrite anion (ONOO^−^) [83]. Peroxynitrite oxidizes almost all biomolecules, including tyrosine-containing proteins, to form the characteristic 3-nitrotyrosine (3-NT). MPO also directly mediates ^●^NO-derived inflammatory oxidants to cause nitration and chlorination in vivo [84].

Evidence suggesting the involvement of MPO in the development of BPD came from clinical observations showing Cl-Tyr [85] and 3-NT [86] were increased in the tracheal aspirates of intubated premature neonates later diagnosed to have BPD. Although increased plasma 3-NT was also reported [87], no changes in plasma Cl-Tyr levels were seen [88]. The discrepancy in plasma and tracheal aspirate Cl-Tyr changes implies that neutrophil activation might be limited in the lungs in the early stage of BPD onset. Other reactive oxidants generated by lung cells include but are not limited to (1) superoxide from NADPH oxidase 2 (NOX2) [89], (2) superoxide from an uncoupled mitochondrial electron transport chain [90], (3) superoxide from xanthine oxidase, (4) hydroxyl radical from the Fenton reaction, (5) endothelial nitric oxidase synthase uncoupling [91], et al. The decreased glutathione metabolism further supports the mechanistic role of OS in BPD [92].

### 6.2. Animal Model OS Studies

The complexity of prematurity-associated morbidities and ethical concerns make studying BPD mechanisms in human subjects almost impossible. Although premature sheep and primates are the best models for studying BPD, their extremely high cost and labor-intensive requirements have prevented most researchers from using them. Rodent pups are born at the canalicular stage and have their alveolar formation started on postnatal day 5 (P5) [26,27]. We have adopted and utilized this animal model for BPD studies. As rodent pups resist OS and counter it by upregulating their antioxidative proteins, an oxygen concentration above 70% is required to generate the BPD phenotype. Using outbred Sprague-Dawley rat pups exposed to >90% O_2_ from P1 to P10, we showed an increased expression of nuclear factor erythroid 2–related factor 2 (NRF2)-mediated antioxidative proteins, and at least one of them had a persistently higher expression than controls even after recovering back to room air (Figure 3). However, this coping mechanism does not prevent the alveolar macrophages from activation and neutrophil recruitment, as evidenced by the significantly increased MPO levels at P4 when morphometric changes are not yet seen in the lungs (Figure 4). Although rat pups can rapidly respond to HOX, alveolar simplification (Figure 5A), neutrophil infiltration, decreasing secondary septation, and a reduction in blood vessel counts (Figure 5B) are seen in the HOX-exposed rat pup lungs at P10, reminiscent of human BPD [93].

### 6.3. Evidence of OS in BPD Lungs

The literature extensively reports evidence of increased OS in BPD [89]. Our studies showed increased levels of Cl-Tyr, 3-NT, and 8-hodroxy-deoxy-guanosine (8-OH-dG), which corroborate increased OS in HOX rat pup lungs (Figure 6) [95]. These findings suggest that despite the rapid upregulation of antioxidative proteins in rat pup lungs, the OS still exceeds the antioxidative capacity, leading to the downstream changes described in the following sections.

### 6.4. Increased ER Stress in BPD Lungs

Evidence of increased ER stress in BPD was first described by the Bhandari group and was described as an integrated stress response [96]. Later, our group reported that early caffeine [93] and tauroursodeoxycholic acid treatments [94] lessened the severity of rat BPD with an attenuation of ER stress. Immunofluorescence detects increased ER stress in autopsied human BPD lungs [94] and BPD rat lungs as early as P4 by immunoblots. Interestingly, the typical markers for ER stress return to control levels after recovering to room air in rat pup lungs. Treating rat pups with one injection of tunicamycin, a potent ER stress inducer, upregulates characteristic ER stress biomarkers in the rat lung, elicits inflammatory cell infiltration, and increases MPO deposition [94]. These results support the mechanistic role of ER stress in BPD, and ER stress reciprocally augments inflammation.

Mitochondria and ER form structural and functional networks essential to cellular homeostasis and metabolism. ER dysfunction will thus impair mitochondrial function, resulting in metabolic dysregulation in BPD lungs [97]. Our metabolomic study shows that BPD lungs rely more on glycolysis to generate ATP with suppressed oxidative phosphorylation. ER also plays a critical role in post-translational protein modification [98]. One characteristic proteomic finding for ER stress is protein dysglycosylation [99]. Several growth factors and their corresponding receptors require glycosylation to reach their maximal activity [100]. A characteristic protein in BPD rat lungs that shows dysglycosylation is the VEGF receptor-2 (VEGFR2, KDR, or FLK1) [94]. We further showed that VEGFR2 dysglycosylation could be seen in tunicamycin-treated rat lungs, and tauroursodeoxycholic acid-treated HOX rat lungs had decreased ER stress with improved VEGFR2 glycosylation. These findings reasonably explain the impaired angiogenesis in the rat BPD model since both VEGF-VEGFR2 signaling [101] and endothelial cell oxidative phosphorylation [102] are critical for successful angiogenesis.

### 6.5. Increased Cellular Senescence in BPD Lungs

The cell responds to DNA damage by (1) activating tumor suppressors to halt DNA replication, (2) recruiting the DNA repair machinery, or (3) initiating programmed cell death if the damage is irreparable [103]. When DNA damage is beyond reparable, the activated tumor suppressor p53 can lead the cell into apoptosis or cellular senescence [104]. An increased binding between p53 and Foxo4 leads DNA-damaged cells to senescence instead of apoptosis [105]. The increased OS and oxidative DNA damage in BPD rat lungs led us to consider whether cellular senescence plays any role in BPD. The increased in situ TUNEL stain, BAX/BCl2 ratio, and cleaved caspase-3 levels indicate increased apoptosis in the BPD rat lungs. Increased ER stress contributes to apoptosis via CHOP (transcriptional factor C/EBP homologous protein)-signaling [106] and caspase-12-signaling (only identified in rodents) [107]. Increased CHOP and cleaved caspase-12 expressions are detected in rat BPD lungs, positively correlating with the extent of apoptosis [93].

When DNA damage is severe enough, the cell might enter a quiescent state that temporarily stops proliferation to allow DNA repair. If the damage is severe enough or persists, the upregulated tumor suppressors can irreversibly shut down the cell cycle, resulting in a senescent state [108]. Senescent cells are hypermetabolic and continuously release growth factors to promote healing and inflammatory mediators to activate sterile inflammation. The secretory activity is called the senescence-associated secretory phenotype (SASP) [109]. Cellular senescence plays multiple biological roles, including organ patterning, which is crucial in fetal organ development (developmental senescence) [110]. A recent study revealed that developmental senescence is essential for lung development before the alveolar stage [111]. Growth and angiogenic factors released as part of the SASP can help organ patterning and promote healing. Senescent cells only account for a minimal percentage of the body in normal conditions and are effectively removed by neighboring phagocytic cells. However, in excessive OS, senescent cells cannot be removed effectively. The accumulated senescent cells thus start to inhibit organ growth by depriving their neighbors of nutrition, converting neighboring cells into senescent cells by the paracrine effect, suppressing the proliferation of stem or progenitor cells, and inducing chronic sterile inflammation via the SASP. We recently reported cellular senescence involving type 2 alveolar cells (AT2), endothelial cells, and several other lung cell types [112]. Since AT2 cells are considered the resident progenitor cells for neonatal lung growth and endothelial cells are the key players in angiogenesis, the senescent change of these two cell types can explain the impaired growth potential of BPD lungs.

We want to draw readers’ attention to the temporal relationship between ER stress and cellular senescence, as ER stress occurs at P4. In contrast, cellular senescence occurs after P4 and persists even after the rat pups have returned to room air for several days [112]. One of the SASP factors is the high mobility group box-1 protein (HMGB1), a potent damage-associated molecular pattern (DAMP) molecule. The release of HMGB1 by senescent cells leads to chronic inflammation in BPD lungs.

## 7. Specific Role of Myeloperoxidase and Sterile Inflammation in BPD

### 7.1. MPO

MPO is a cationic heterotetrameric hemoprotein (~150 Kd) composed of two light chains (~15 Kd) and two extensively glycosylated heavy chains (~64 Kd). The light chains contain the modified iron protoporphyrin IX active site. The heme core can have several redox states that play a critical role in eight reactions due to its iron center, a transitional metal [113] (Figure 7). Macrophages and neutrophils are the two types of white cells that contain MPO. MPO can be released from myeloid cells without infection [114]. In the presence of hydrogen peroxide, MPO converts halide (Cl^−^, Br^−^, or I^−^) or pseudohalide (e.g., SCN^−^) into reactive oxidants (HOCl, HOBr, HOI, or HOSCN) through its peroxidase cycle (reaction 1) and halogenation cycle (reaction 2) to kill microorganisms [115]. MPO can also generate superoxide and hydrogen peroxide through a superoxide-induced cycle or augment the peroxynitrite-induced damage [116]. MPO has catalase activity (reaction 5) that can be repurposed to reduce OS [117].

Although MPO is considered a critical protein for innate immunity, intriguingly, its depletion does not seem to increase the susceptibility to infections unless superimposed with poorly controlled diabetes mellitus [118]. Animal studies have revealed that NADPH oxidase, but not MPO, might be more critical for animals to fend off infections [119]. The contributory role of MPO in BPD came from the observation that the levels of Cl-Tyr were significantly higher in the tracheal aspirates of premature neonates who were later diagnosed with BPD [85]. The rationale is that MPO is the only biological source for Cl-Tyr [120]. Increased MPO expression in BPD rat lungs at P4 (Figure 4) corroborates human findings and suggests MPO plays a critical role in BPD onset.

### 7.2. Other Inflammation-Related Proteins in BPD Lungs

Our rat BPD model showed that HOX-induced OS initiated an MPO-mediated sterile inflammatory response in the lungs. After the initial response, we observed several inflammatory mediators being upregulated, including high mobility group box 1 (HMGB1), toll-like receptor 4 (TLR4), the receptor for advanced glycation end-products (RAGE) [95], and *S*-nitrosoglutathione reductase (GSNOR) (preliminary data).

a.HMGB1

HMGB1 is a non-histone nucleoprotein that organizes DNA and regulates transcription factors when it stays in the nucleus. Structurally, HMGB1 consists of three discrete domains: the A-box, B-box, and an acidic C-terminal. When a cell sustains an injury, HMGB1 will be released into the cytosol and become acetylated before being released into the extracellular space. However, if the cell dies from the insult, HMGB1 will be released without being acetylated. The non-acetylated HMGB1 is thus related to the extent of cell death [121]. Once in the extracellular space, HMGB1 becomes a potent damage-associated molecular pattern (DAMP) molecule that binds to several receptors to activate the transcription factor nuclear factor kappa light-chain enhancer of B-cells (NFkB), signaling downstream inflammation [121]. HMGB1 contains three thiols and can form disulfide bonds with other thiol-containing molecules or form intramolecular disulfide bonds. HMGB1 can be pro-inflammatory or anti-inflammatory depending upon its redox state and disulfide formation [122]. The fully oxidized (ox-) and reduced HMGB1 are anti-inflammatory, whereas the disulfide HMGB1 is pro-inflammatory. The level of HMGB1 increases very early at P4 in BPD rat lungs (Figure 8) and remains high at P10 (Figure 9A). HMGB1 binds to several receptors to activate NFκB signaling and downstream inflammatory responses. TLR4 and RAGE are the essential receptors mediating HMGB1 activity [123].

b.TLR4

TLR4 is a pattern recognition receptor with lipopolysaccharide (LPS) as its primary ligand. TLR4 also recognizes endogenous DAMP molecules such as HMGB1 in non-infectious conditions. Two pathways are activated independently after the ligand binds to TLR4—myeloid differentiation primary response 88 (MyD88)-dependent and TIR-domain-containing adapter-inducing interferon-β (TRIF)-dependent pathways [124]. HMGB1 uses its A-box to bind TLR4 with the help of myeloid differentiation protein 2 (MD2) protein to activate NFB signaling [125]. Reagents that inhibit MD2 can thus serve as inhibitors for the HMGB1-TLR4 signaling pathway. Interestingly, the binding between HMGB1 and TLR4 will upregulate the surface expression of TLR4 [123]. Although we did not see any change in TLR4 levels at P4 (Figure 8), 2.5-fold levels of TLR4 were seen in BPD lungs at P10 (Figure 9B), indicating that the increased expression occurs later than the HMGB1 surge.

c.RAGE

RAGE is also a pattern recognition receptor with a single transmembrane domain that can bind multiple ligands to signal inflammation, proliferation, apoptosis, autophagy, and cell migration [126]. Cell expression is relatively low except for alveolar type I (AT1) cells, which are crucial for maintaining lung structure. The high expression of RAGE on AT1 cells may be one reason acute lung injury is frequently seen in sepsis or other organ inflammation. HMGB1 binds to the full-length RAGE on the cell membrane to activate diaphanous homolog 1 (Diaph1), toll-interleukin-1 receptor domain-containing adaptor protein (TIRAP), or MyD88 signaling, followed by activation of NFkB or the type 1 interferon pathway [127]. This results in a pro-inflammatory response, including M1 macrophage polarization. The expression of RAGE in BPD rat lungs is similar to TLR4 (Figure 9C) and corroborates the report that HMGB1 binds to RAGE [123]. A few truncated RAGE isoforms can be generated via either mRNA alternative splicing or by enzymatic cleavage from the cell membrane and released extracellularly. These truncated isoforms can be grouped as soluble RAGE. Soluble RAGE can act like a RAGE inhibitor by binding HMGB1 without activating downstream signaling [128].

d.GSNOR

GSNOR, or alcohol dehydrogenase 5 (ADH5), breaks down *S*-glutathione to decrease the bioavailability of ^●^NO [129]. This enzyme can protect animals from septic shock by reducing pathological ^●^NO production [130]. A physiological amount of ^●^NO is essential for vasomotor regulation [131] and mitochondrial biogenesis [132]; both latter processes are critical for neonatal lung growth. Raffay and his group first reported the mechanistic role of GSNOR in BPD. They demonstrated that global GSNOR knockout protected mice from BPD, while smooth muscle cell-specific GSNOR had no effects [133]. Our HOX rat BPD model detected an increased GSNOR expression at P10 (Figure 10A). Providing a GSNOR inhibitor, N6022 [134], during HOX exposure effectively attenuated the morphometric changes in the lungs (Figure 10B).

The mechanism by which GSNOR is upregulated is probably through NFκB signaling [135]. As previously described, HMGB1 binding to TLR4 or RAGE activates NFκB signaling, possibly leading to GSNOR overexpression. Our sickle cell disease study demonstrated that HMGB1 contributes to increased GSNOR activity and lung damage [136]. Thus, we hypothesized that HMGB1 could upregulate GSNOR expression. By culturing three lung cell types with endotoxin-depleted recombinant HMGB1 for 48 h in vitro, the expression of GSNOR increased significantly (Figure 11A). Similar treatment also improves the biomarkers’ expression for ER stress (Figure 11B) and cellular senescence (Figure 11C). These findings suggest HMGB1 is upstream of GSNOR, ER stress, and cellular senescence.

### 7.3. The BPD Sterile Inflammatory Pathway

Our study results suggest that at least four sources of OS contribute to BPD onset and progression. The first source directly comes from HOX, which activates alveolar macrophages and subsequent neutrophil recruitment. The MPO released by the activated neutrophils uses H_2_O_2_ produced by the activated macrophages and the chloride anion in the surrounding biofluid to generate HOCl as the second source of OS. The amplified OS kills or damages cells to release HMGB1 or disturbs ER function, leading to ER stress. Cells gear up the unfolded protein response to resume ER function, which inadvertently leads to the third source of OS [137]. ER stress disturbs cellular homeostasis, and the massive amount of OS locally can damage DNA or activate inflammation [138] to cause cellular senescence [112]. The combined effect of metabolic dysregulation, hypermetabolism, and inflammation from cellular senescence provides the fourth source of OS.

Our studies also identified the critical involvement of HMGB1 signaling [95], ER stress [93,94], and cellular senescence in BPD [112]. These signaling pathways interact with each other in a very complex fashion (Figure 12). HMGB1 released in the early stage of BPD onset binds to TLR4 and RAGE to initiate subsequent inflammatory reactions contributing to ER stress and cellular senescence. ER stress contributes to mitochondrial electron transport chain uncoupling, which encourages cellular senescence. ER stress also augments sterile inflammation by upregulating cyclo-oxygenase activity [93,96]. The SASP from cellular senescence reciprocally amplifies inflammation by releasing inflammatory mediators, including HMGB1. The impaired glycosylation of VEGFR2 and mitochondrial dysfunction from ER stress, AT2 and endothelial cells’ depletion from cellular senescence, and deprived ^●^NO bioavailability from increased GSNOR activity all contribute to the impaired growth trajectory of BPD lungs. Based on these data and previously published studies, we believe these processes constitute a BPD-sterile inflammatory pathway, as summarized in Figure 12.

## 8. Targeted Therapeutic Intervention

The sterile inflammation pathway outlined in Figure 12 and discussed in the previous section offers new insights into the onset and progression of BPD. This provides a promising opportunity for developing a targeted therapeutic approach to treat this complex disease safely and effectively. In particular, we would like to raise the question of whether simple targeting of MPO provides a practical approach.

### MPO Inhibitors

Our data demonstrate that MPO-mediated OS plays a crucial role in BPD onset and progression [94]. As HOX-induced OS seems to be the first step in activating alveolar macrophages, it is reasonable to consider using antioxidants to prevent BPD. Previous studies have demonstrated the efficacy of exogenous administration of vitamins and micronutrients in reducing OS. Nevertheless, the encouraging preclinical results failed to translate into successful clinical efficacy [139]. As MPO is a specific source that amplifies OS in BPD lungs, we hypothesized that inhibiting MPO activity could offer a benefit. It should be noted that MPO does not appear to be an essential or critical enzyme in human physiology [118], and its depletion does not seem to increase the susceptibility to infections.

a.Irreversible MPO inhibitor

Almost all inhibitors reported in the literature inhibit the formation of HOCl irreversibly [140]. Azide has traditionally been used as the traditional MPO inhibitor, but its use has been superseded by 4-aminobenzoic acid hydrazide (ABAH). ABAH is a suitable one-electron donor for compound I, facilitates compound II formations, and prevents compound I from being reduced to the resting state (halogenation cycle, cycle 2 in Figure 7). ABAH is also a poor electron donor for compound II, which prevents compound II from being reduced to the resting state (peroxidase cycle, cycle 1 in Figure 7) [141]. Poor palatability and a short half-life are two main problems that make ABAH incompatible for clinical use. Thioxanthine derivative AZD5904, thiouracil derivative PF-06282999, and two unique molecules, AZD3241 and AZD4831, belong to this group. Only one compound, AZD4831, entered a Phase 3 clinical trial in congestive heart failure patients with relatively preserved ejection fraction. The long half-life makes AZD4831 a promising MPO inhibitor.

b.Reversible MPO inhibitor

Natural compounds, such as melatonin and tryptophan, reversibly inhibit the MPO catalytic cycle. Melatonin acts as an electron donor for compound I, accelerating compound II formation and converting MPO peroxidase activity to catalase-like activity. The accumulation of compound II will prevent MPO cycling. Some indole derivatives show similar activity and inhibit MPO reversibly, effectively competing with Cl^−^ and SCN^−^ to prevent compound I from generating HOCl and hypothiocyanate, identical to melatonin [142]. Some flavonoids and polyphenols have reversible MPO inhibitory activities, such as quercetin, the most studied natural compound and a potent natural MPO inhibitor [143]. Unfortunately, the mechanism by which these natural compounds inhibit MPO activity has not been thoroughly investigated.

It is known that MPO preferentially oxidizes the phenolic ring of tyrosine in small peptides to form reactive oxidants [144]. This preference is mainly due to the structure of tyrosine, which can adequately dock itself to the heme pocket of MPO. The property makes tyrosine an ideal competitor to prevent hydrogen peroxide from interacting with halide or pseudohalide. Our group hypothesized that a cysteine next to tyrosine can scavenge ^●^Tyr from MPO by its thiol group. After extensive in vitro experiments on tripeptides, we identified that lysyltyrosylcysteine was MPO’s most potent candidate for decreasing HOCl generation, with an IC_50_ of ~7 µM [145]. Its MPO inhibitory effect is through its binding to compounds I and II. To improve its stability, we end-capped the tripeptide to obtain N-acetyl-lysyltyrosylcysteine amide (KYC). Animal studies have shown KYC effectively attenuates the severity of limb ischemia in diabetic mice [146], vasoconstriction in sickle cell disease [147], autoimmune encephalomyelitis [148], plaque psoriasis [149], stroke [150], lung damage in sickle cell disease [136], and BPD [95]. Our pharmacokinetic study showed a short T_1/2_ (~60 min in our preliminary research using rat pups at P10) with prolonged biological efficacy that inhibited MPO’s activity reversibly. No toxicity to endothelial cells was seen at a concentration of 4000 μM.

c.MPO inhibitors in BPD

The crucial role of inflammatory cells in BPD has been demonstrated for decades. The finding that increased Cl-Tyr in tracheal aspirates [85] indicated that MPO might be a potential therapeutic target for BPD. The increased MPO as early as P4 [94] and Cl-Tyr in BPD lungs at P10 [95] undoubtedly support the importance of MPO in BPD onset and progression. Our group showed that KYC protects animals from several MPO-mediated disorders, so we determined if MPO inhibition is a viable therapeutic strategy to decrease BPD. KYC and AZD4831 intraperitoneal injections were adopted to represent reversible and irreversible MPO inhibitors. As reported in the literature, KYC was given at a daily dose of 10 mg/kg [95], and AZD4831 was given every other day at 1 5 mg/kg [151].

In our studies, KYC consistently improves alveolar complexity (Figure 13A) and increases blood vessel density and the secondary septation of BPD rat lungs [94,95,112]. Based on the reproducibility of the findings, we conclude that KYC attenuates HOX-induced BPD. The decreased MPO(+) cell infiltration (Figure 13B) and MPO distribution (Figure 13C) confirm the decreased MPO expression level we previously reported [95]. The decreased MPO and Cl-Tyr levels are parallel to the reduced apoptosis (Figure 13D), ER stress (Figure 13E), and cellular senescence (Figure 13F). These results strongly indicate that KYC protects HOX-exposed rat pup lungs. Combined with our findings from AZD4831-treated BPD lungs described in the next section, we realize that the benefit is not due to simple MPO inhibition alone.

The second mechanism by which KYC attenuates BPD is by modulating inflammatory signaling. As previously described, HMGB1, TLR4, and RAGE expressions increased in BPD rat lungs; KYC treatment significantly decreased these levels (Figure 14A). We also detected reduced bindings between HMGB1 and TLR4/RAGE (Figure 14B). KYC facilitates HMGB1 oxidation and acetylation (Figure 14C) and prevents ds-HMGB1 formation by thiylating the sulfhydryl groups of HMGB1. Ox-HMGB1 is the anti-inflammatory isoform; the increased ox-HMGB1 isoform in BPD lungs thus provides another explanation for the KYC’s beneficial effects. Acetylated (Ac-) HMGB1 is released from activated myeloid cells, as Ac-HMGB1 is the major isoform in KYC-treated BPD lungs. This finding indicates that cell death is decreased in KYC-treated BPD lungs.

The third mechanism is by upregulating antioxidative capacity, probably due to the increased expression of Nrf2. The increased binding between Keap-1 and Nrf2 in BPD lungs encourages ubiquitin-proteosome-mediated Nrf2 breakdown [95]. KYC decreases this Keap-1-NRF2 binding through thiylation and glutathionylation of Keap-1, thus allowing Nrf2 to escape degradation [95]. This efficacy occurs in BPD and normal control lungs (Figure 14D). We used the RLMVEC in vitro system to study how KYC achieved Nrf2 upregulation and were surprised to see that MPO is needed to obtain Nrf2 upregulation. Our data show that KYC protects the neonatal lungs against BPD through at least three mechanisms, indicating that KYC is a systems pharmacology drug [152].

In contrast to KYC, AZD4831 did not improve the morphometrics of BPD lungs (Figure 15). Although we have not extensively studied AZD4831 compared with KYC, the lack of improvement indicates that a suicidal MPO inhibitor is probably not an optimal option for BPD treatment. This finding also suggests that repurposing MPO into quasi-catalase (cycle 5 of Figure 7), making KYC a systems pharmacology agent, is the primary mechanism for KYC’s many layers of protection against BPD.

## 9. Discussion

The standard of care in Neonatal Intensive Care Units for premature neonates has improved significantly over the past decade. This has led to a decreased mortality rate in this fragile patient population. The improved survival of those extremely premature neonates also contributes to the unchanged or even slowly increased incidence of BPD. The annual incidence of BPD in the USA remains mostly unchanged at ~15,000 patients per year [1,3,4]. This suggests a limited understanding of this highly complex and variable disease condition.

Several socioeconomic issues negatively influence the development of new BPD treatments. Although the US Congress passed the “Best Pharmaceuticals for Children Act” in 2002 and the “FDA Safety and Innovation Act” in 2012 to encourage funding agencies to sponsor clinical trials for off-patent drugs in pediatric patients, new drug trials in neonates remain difficult. One main reason is that studies on neonates take a long time, which means the cost will be much higher than that of adults. Researchers would need to follow enrollees for years, requiring more human and financial resources to monitor the studies. The narrow profit margin discourages pharmaceutical companies from investing in such studies unless the government is willing to provide reasonable incentives. Researchers developing neonate-centered therapies require well-prepared plans with clear sustainable development goals included before embarking on such studies. This unique and complicated situation is an obstacle in the path to a successful new drug investigation for neonates.

As highlighted in the current work, the actual definition of BPD continues to change and evolve (see Table 1 and Section 2 above). Although many risk factors have been reported in the literature, no one can be used quickly and confidently to predict which premature neonates will develop BPD and deserve more attention or need to be treated differently. Some reported risk factors even have contradictory impacts on BPD in different studies, such as chorioamnionitis. Low GA remains the most critical risk factor for BPD because they are born before the alveolar stage, which we have no way to rectify. Criteria for the diagnosis of BPD continue to be somewhat rudimentary and depend on the patient’s need for supplemental oxygen. No biomarkers and/or diagnostic markers are currently used in the BPD population (see Section 5 above). The understanding of mechanisms associated with BPD onset, progression, and therapeutic intervention is exceedingly limited (see Section 6 and Section 7 above). Finally, the treatment of BPD is also somewhat limited. There are no FDA/EMA-approved drugs specifically for BPD (see Section 8 above).

Vitamin A and caffeine are the only measures with clinically proven efficacy in reducing BPD. Caffeine is the only BPD treatment widely accepted by clinicians. Interestingly, the mechanism behind BPD prevention by caffeine is unknown. One plausible mechanism for caffeine’s effect is the attenuation of ER stress [93]. However, after almost two decades of the Caffeine Therapy for Apnea of Prematurity (CAP) trial [24], there seems to be no change in BPD prevalence. New therapeutic interventions are required to improve the long-term pulmonary outcome of premature neonates. Numerous mechanisms have been studied that may cause the BPD phenotype, and thus far, they have provided limited insights that translate into specific and effective therapies.

Using the HOX rat BPD model, our group has gained insight into previously under-investigated mechanisms for BPD. The significantly increased Cl-Tyr in tracheal aspirates [85] of premature neonates, but not in blood [88], who later developed BPD, led our group to hypothesize that MPO in HOX-exposed neonatal lungs might play a crucial contributing role in BPD. The early upregulation of MPO and HMGB1 in the saccular stage of rat lungs under HOX provided an excellent reason to investigate this pathway further. Through our studies, we have identified that MPO-induced oxidative stress [93], ER stress [94], and cellular senescence [112] contribute to the onset and progression of BPD. We also identified that MPO-mediated OS is intimately associated with HMGB1-TLR4/RAGE signaling [95]. Using KYC as a probe, we have also linked MPO-mediated OS to ER stress and cellular senescence in BPD lungs.

ER stress impairs angiogenesis by decreasing the N-glycosylation of angiogenic factors and their cognate receptors, uncoupling the mitochondrial electron transport chain, and depriving the availability of ^●^NO [94]. As angiogenesis plays a crucial role in alveolar formation, its impairment reasonably explains the development of BPD. The alveolar formation also needs alveolar cells to work cooperatively with endothelial cells. AT2 has been considered the progenitor cell for alveolar formation among the two types of alveolar epithelial cells. The identification of a senescent change of AT2 in BPD rat lungs provided another explanation for the poor lung growth trajectory [112]. Based on all our findings, we have constructed the “BPD sterile inflammatory pathway” to highlight the intricate and complicated interactions between the key protein constituents and the associated feedback loops (Figure 12).

We have previously discussed the need to develop systems pharmacology drugs to effectively treat complex disease conditions such as BPD [152]. A systems pharmacology drug typically interacts with more than one target in a pathway or network. The more common “one drug—one target” has approached significant limitations for complex disease states. For example, in our current work described here, MPO has been shown to attenuate HMGB1-mediated inflammation, ER stress, and cellular senescence. However, simply targeting MPO with a well-characterized one-drug-one target inhibitor such as AZD4831 in our rat pup model of BPD failed to show any phenotype improvements, as shown in Figure 15. In comparison, our systems pharmacology drug candidate KYC effectively alleviates the well-described phenotype endpoints in the rat pup model of BPD.

KYC is a first-in-class systems chemico-pharmacology drug (SCPD) that exhibits promise in treating BPD. The physicochemical and pharmacological properties of such a drug are as follows:i.The SCPD (KYC) interacts with a first target that comprises a druggable biomolecular site, in this case, MPO.ii.The SCPD (KYC) modifies the properties and functions of the first target (MPO—inhibits the production of toxic oxidants and converts to a quasi-catalase) and is itself chemically modified (forms an initial tyrosyl radical, which through intramolecular transfer to cysteine forms a thiyl radical) and further activated by the first target.iii.The chemically modified and activated SCPD (KYC) interacts with one or more secondary targets (HMGB1, KEAP-1 Nrf2), thus modifying each of the second targets’ activity and functions.

The activation of KYC by MPO creates a reactive thiyl radical species that can react with proximal components involved in the inflammation process occurring in BPD onset and progression. The reaction of the KYC-thiyl radical appears to react and shut down key elements of the cascade process, such as HMGB-1. However, because the half-life of the thiyl radical of KYC is limited (micro-milliseconds), the reactions that can occur are limited in scope by proximity. We refer to this as the “Sphere of Influence” for KYC reactions. This means that KYC is activated only when MPO is present, and the reactivity of the thiyl radical of KYC is determined by the Sphere of Influence. Therefore, reactions are limited to the area of inflammation where MPO is present, and off-target reactions are limited.

## 10. Conclusions

BPD is a complex, multifactorial pulmonary complication that occurs mainly in premature neonates. Treatment for BPD that focuses only on a single target in the pathways/network associated with BPD is likely to fail due to the complexity of this lung problem. By modulating MPO, KYC offers different protection against BPD as a SCPD agent. However, the study of this novel peptide remains in its early stages. As inflammation is another critical contributor to BPD, future animal studies should investigate whether KYC offers similar protection against endotoxin-induced BPD. Before clinical studies, we also need to understand (1) whether KYC provides similar protection against BPD in other animals, (2) the pharmacodynamics, pharmacokinetics, and toxicology of KYC, and (3) the proper routes of administration. Our findings suggest that SCPD should be the best kind of therapeutic approach for complex disorders. The scientific community should consider the potential of SCPDs and their putative value in the treatment of BPD and other complex disease states.

## Figures and Tables

**Figure 1 antioxidants-13-00889-f001:**
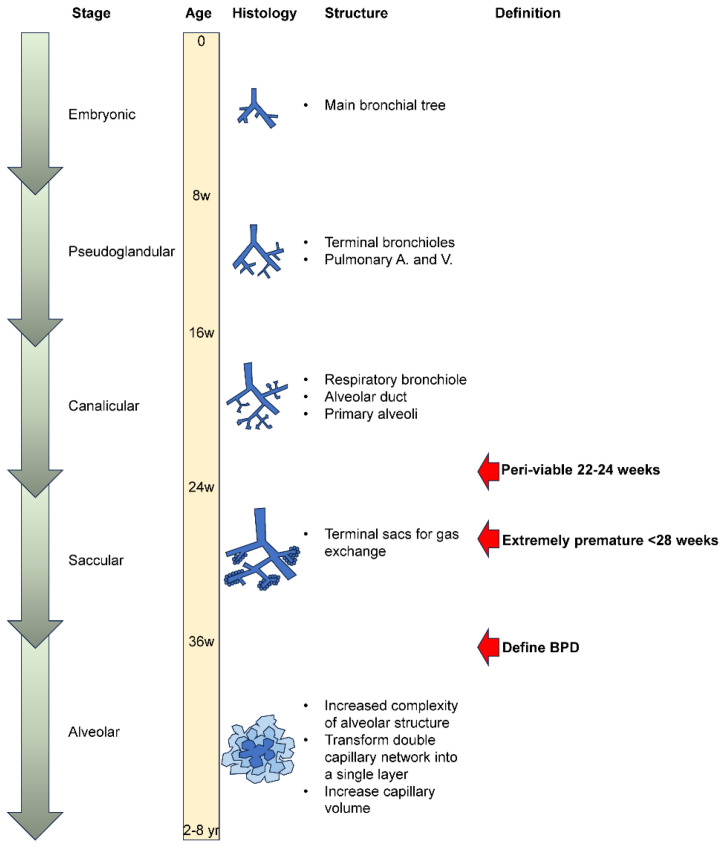
Developmental stages of the lungs. Lung development can be divided into five stages, including embryonic (0–8 weeks), pseudoglandular (8–16 weeks of gestation), canalicular (16–24 weeks of gestation), saccular (24–36 weeks of gestation), and alveolar (36 weeks of gestational to 2–8 years of age). Biologically, neonates born after 36 weeks of gestation should be able to breathe without treatment or support. The oxygen requirement after 36 weeks post-conceptionally is thus chosen as the definition for BPD for extremely premature infants born before 32 weeks.

**Figure 2 antioxidants-13-00889-f002:**
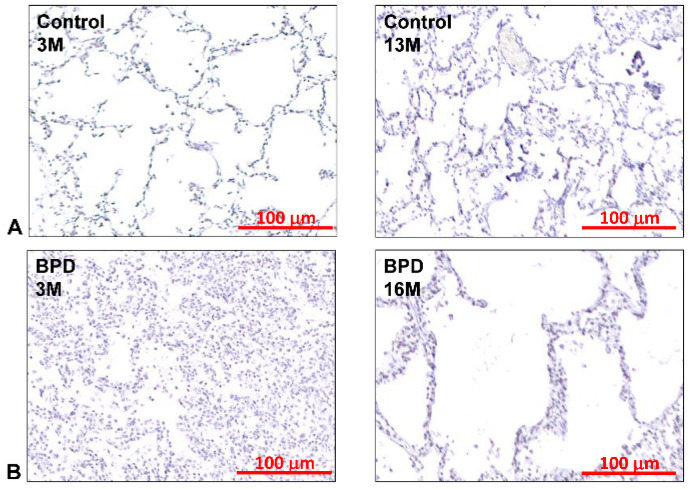
Pathologic findings of human BPD. As compared to the thin-wall and well-septated alveoli in normal lungs (**A**), BPD lungs are characterized by alveolar simplification, decreased septation, thickened alveolar walls, decreased capillary counts, alveolar hyperinflation mixed with collapse, and superimposed inflammatory cell infiltration (**B**). The pathologic findings reasonably explain the lung stiffness, poor oxygenation, restricted airway, and thickened secretion clinicians encountered during the care of BPD infants.

**Figure 3 antioxidants-13-00889-f003:**
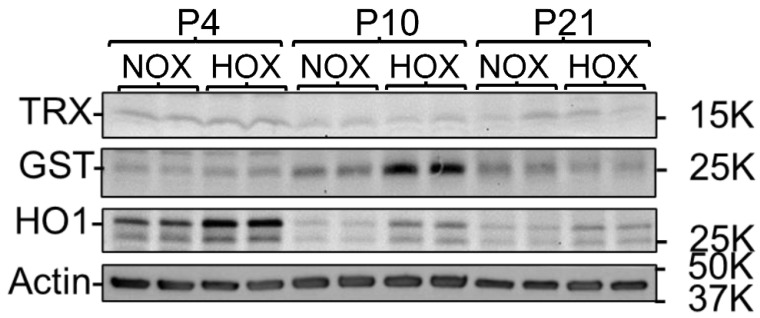
Rat pups upregulate lung antioxidative proteins after exposure to HOX. Rat pups can upregulate the nuclear factor erythroid 2-related factor 2 (Nrf2)-mediated antioxidative enzyme expression. Rat pups were exposed to >90% O_2_ (HOX) from postnatal day 1 (P1) to P10 and then recovered to room air afterward. Three representative Nrf2-mediated proteins—thioredoxin-1 (TRX), glutathione-*S*-transferase (GST), and heme-oxygenase-1 (HO1)—in the lungs were quantified by immunoblots. All three proteins are upregulated as early as P4. The increased GST and HO1 expressions persisted throughout HOX exposure. The upregulated HO1 persisted even after the pups recovered to room air for 11 days (P21). (Reprinted from reference [94] with permission).

**Figure 4 antioxidants-13-00889-f004:**
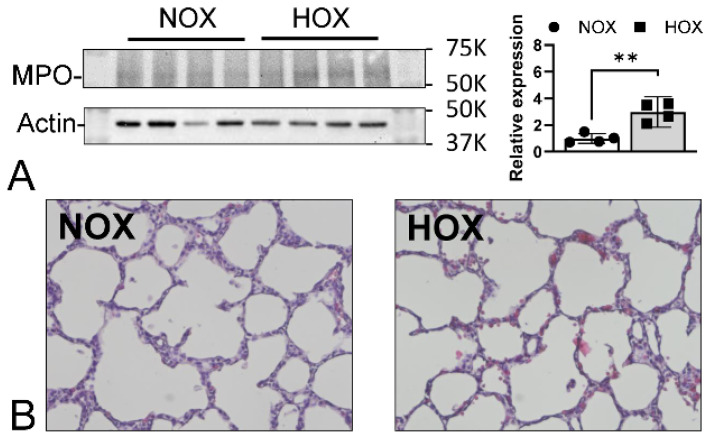
MPO contributes to BPD onset. HOX activates alveolar macrophages to recruit the circulating neutrophils, as evidenced by increased lung MPO expression at P4 (**A**). Neutrophil infiltration might contribute to BPD onset, as morphometric BPD changes are not seen at this point (**B**). **: *p* < 0.01 (n = 4).

**Figure 5 antioxidants-13-00889-f005:**
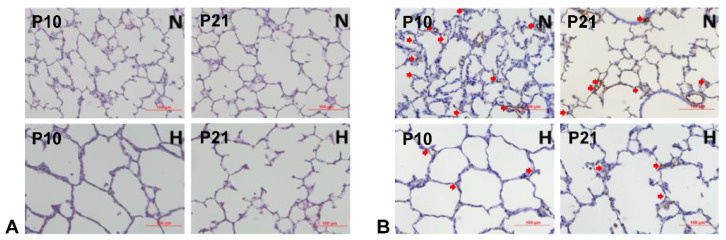
HOX exposure causes the BPD phenotype in rat pups. Alveolar simplification, reductions in radial alveolar counts and secondary septations, and infiltration of inflammatory cells are seen in the lungs under H&E stains (**A**). The decreased blood vessel counts in the HOX-exposed lungs indicate impaired angiogenesis under the immunohistochemistry stains using an antibody against the rat endothelial cell antigen (**B**). N: NOX, H: HOX. Red arrow: endothelial cells in blood vessels. Scale bar = 100 μm.

**Figure 6 antioxidants-13-00889-f006:**
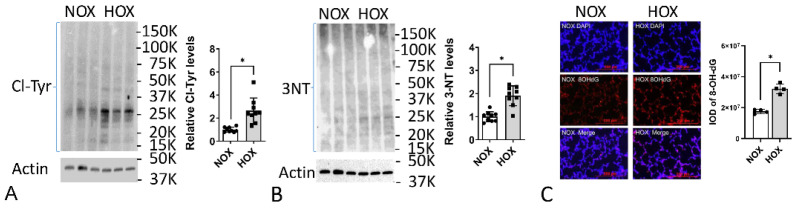
OS is increased in BPD rat lungs. (**A**) The levels of Cl-Tyr are 2.6-fold higher in BPD lung lysates, indicating an HOCl-mediated OS. (**B**) The levels of 3-NT are 1.9-fold higher in BPD lung lysates, indicating a peroxynitrite- or reactive nitrogen species-mediated OS. (**C**) The integrated signals of 8-OH-dG are 1.8-fold higher in BPD lung, indicating OS-mediated DNA damage. (Reprinted from reference [95]. Used with permission). Scale bar = 200 μm. *: *p* < 0.05 (n = 12).

**Figure 7 antioxidants-13-00889-f007:**
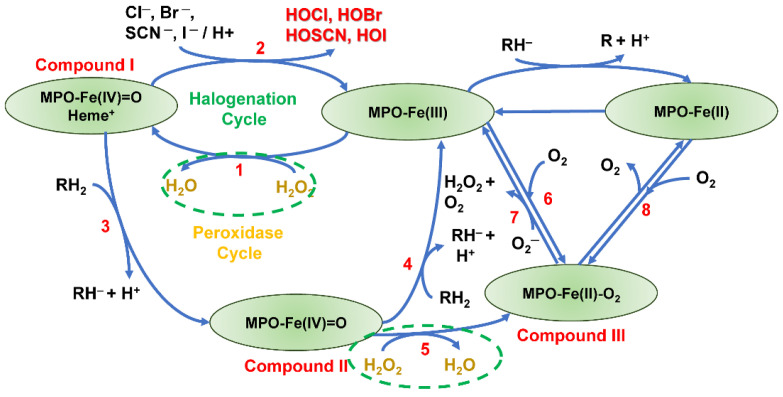
Summary of the MPO activities. MPO comprises two heavy chains and two light chains with a heme core. Iron can have multiple redox states as a transition metal; hence, there are three compounds for MPO. Through the halogenation and peroxidase cycles, compound I converts the chloride anion into hypochlorous acid, where hydrogen peroxide is available (reaction 2). There are two conditions in which MPO functions as catalase (reactions 1 and 5; green ovals with dotted outlines).

**Figure 8 antioxidants-13-00889-f008:**
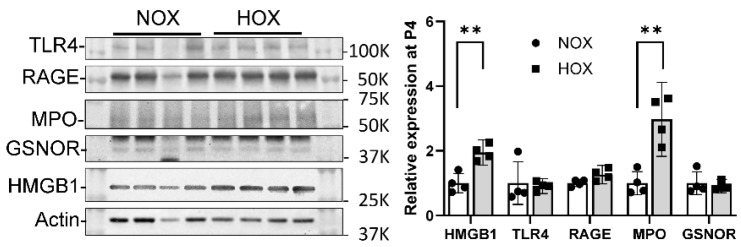
Early expression of inflammatory proteins in the BPD rat lungs. The HMGB1 levels significantly increase at P4, when MPO levels also increase. The TLR4, RAGE, and GSNOR levels do not show a significant change. This finding suggests both HMGB1 and MPO are involved in BPD onset. **: *p* < 0.01 (n = 4).

**Figure 9 antioxidants-13-00889-f009:**
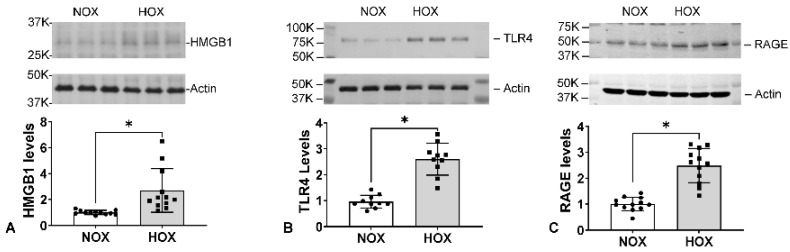
Increased expression of inflammatory proteins during BPD progression. Under HOX exposure, the expressions of HMGB1 (**A**), TLR4 (**B**), and RAGE (**C**) are all increased at P10. (Reprinted from reference [95] with permission). *: *p* < 0.05 (n = 12).

**Figure 10 antioxidants-13-00889-f010:**
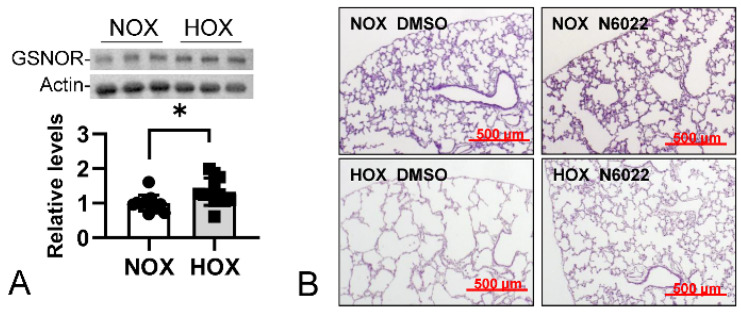
GSNOR contributes to the BPD progression. (**A**) GSNOR expression increased by 30% in HOX rat BPD lungs. (**B**) Daily treatment with N6022, a GSNOR inhibitor, effectively attenuates the BPD changes in HOX-exposed rat lungs, suggesting a contributory role of GSNOR in BPD progression. *: *p* < 0.05 (n = 12).

**Figure 11 antioxidants-13-00889-f011:**
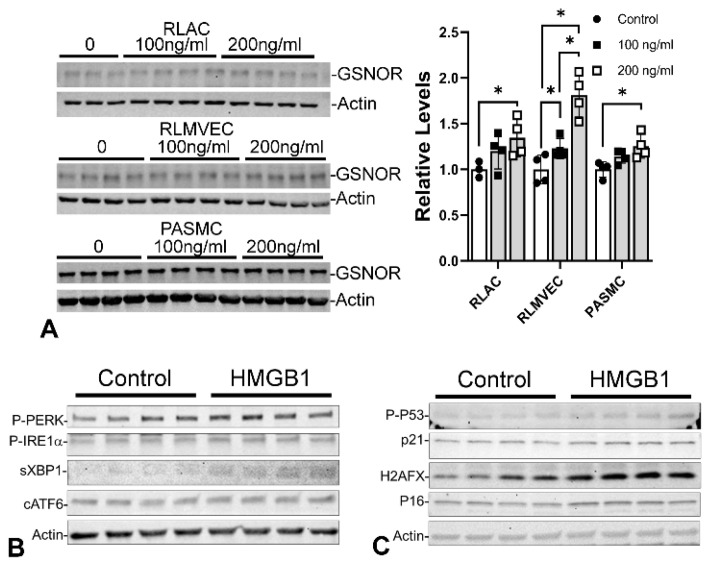
HMGB1 in vitro treatment upregulates GSNOR expression, ER stress markers, and cellular senescence markers. (**A**) Two concentrations (100 ng/mL and 200 ng/mL) are used to treat RALC, RLMVEC, and PASMC for 48 h. Immunoblots quantify the expression of GSNOR. GSNOR expression increases in all three lung cell types, with RLMVEC having the highest response. (**B**) Changes in ER stress are studied in RLMVEC. All ER stress markers increase after 200 ng/mL HMGB1 treatment for 48 h. (**C**) Change in cellular senescence is also studied in RLMVEC with 200 ng/mL HMGB1 treatment for 48 h. The commonly used markers for cellular senescence increase after HMGB1 treatment. RLAC: rat lung alveolar cells; RLMVEC: rat lung microvascular endothelial cells; PASMC: pulmonary artery smooth muscle cells from rat pups. * *p* < 0.05 (n = 3~4).

**Figure 12 antioxidants-13-00889-f012:**
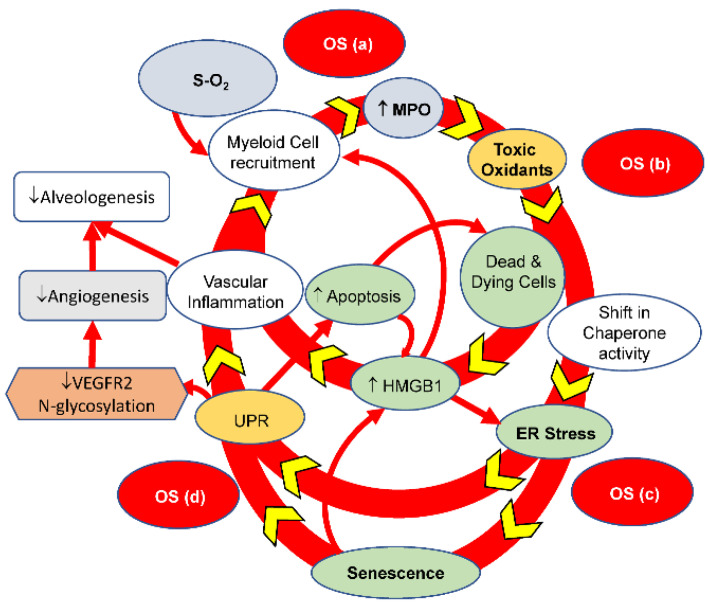
The BPD-sterile inflammatory pathway. Exposing rat pups to hyperoxia elicits oxidative stress-mediated lung responses that culminate in the BPD phenotype. The figure summarizes our recent findings. These responses interact with each other to form a complex relationship. There are at least four sources of OS, including hyperoxia (a), myeloperoxidase (b), endoplasmic reticulum stress (c), and cellular senescence (d). The OS coming from HOX can be reduced clinically, to a limited extent, through the judicious use of oxygen. Specific inhibitors can target the other three OSs. However, MPO-mediated OS seems upstream of the other two and could be the best therapeutic target. This is a newly edited figure with cellular senescence included. (Reprinted from reference [94] with permission). ↑: increase; ↓: decrease.

**Figure 13 antioxidants-13-00889-f013:**
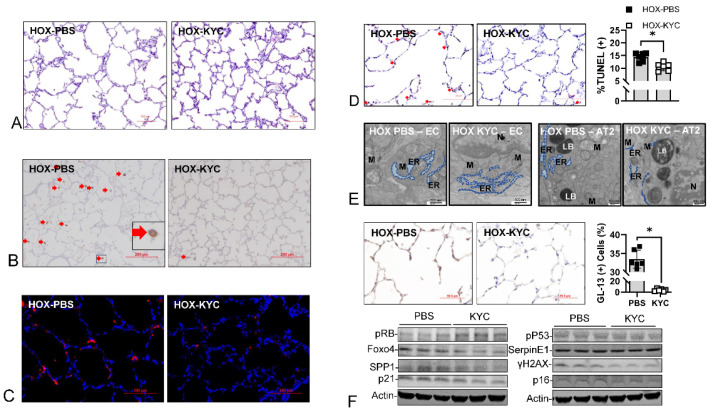
KYC protection of HOX-exposed lungs. KYC daily intraperitoneal injection of 10 mg/kg improves the alveolar formation (**A**), decreases MPO(+) inflammatory cell infiltration (red arrows) (**B**), and immunofluorescence stain of MPO (red) distribution (**C**), apoptosis (**D**), ER stress (**E**), and cellular senescence (**F**). The MPO immunofluorescence (red) stain shows reduced MPO release into the tissue. The nuclei are stained with DAPI (blue). The in situ TUNEL stain represents DNA damage and is used to estimate apoptosis. The dilated ER structure under the electron microscope indicates that KYC attenuates ER stress. GL-13 stain, a modified lipofuscin stain equivalent to the acidic β-galactosidase activity stain, and commonly used markers (p16, p21, p53, γH2AX or H2AFX, SerpinE1, and SPP1) are used to quantify cellular senescence. LB: lamella body; M: mitochondria (Reprinted from references [94,112] with permission). (**A**,**C**,**D**) Scale bar = 100 μm; (**B**) Scale bar = 200 μm; (**E**) Scale bar = 500 μm; (**F**) Scale bar = 50 μm. *: *p* < 0.05 (n = 6).

**Figure 14 antioxidants-13-00889-f014:**
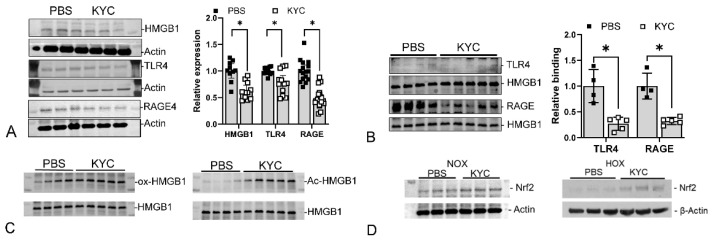
KYC attenuates inflammatory signaling and augments antioxidative capacity in HOX-exposed lungs. (**A**) The major DAMP protein HMGB1 and its two main receptors, TLR4 and RAGE, decrease in BPD lungs after KYC treatment. (**B**) Immunoprecipitation of the BPD lung lysates by HMGB1 antibody shows decreased bindings of TLR4 and RAGE with HMGB1 on immunoblots by KYC. (**C**) Using the same immunoprecipitation strategy, we see oxidized (ox-) and acetylated (Ac-) HMGB1 increases in BPD lungs by KYC. (**D**) Although the increased Nrf2 expression in room air-exposed (NOX, **left** panel) lungs is not as dramatic in HOX-exposed (**right** panel) lungs, KYC increases the antioxidative capacity of neonatal lungs by upregulating Nrf2 expression in both oxygen environments. (Reprinted from reference [95] with permission). *: *p* < 0.05, (n = 12 and 4 for (**A**) and (**B**), respectively).

**Figure 15 antioxidants-13-00889-f015:**
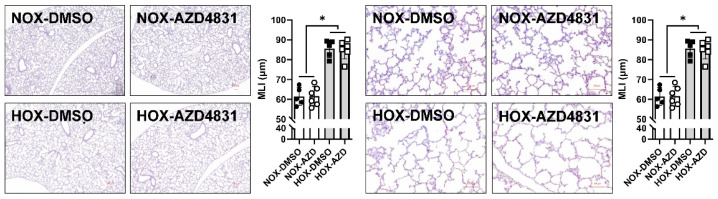
Irreversible MPO inhibition does not attenuate BPD. Rat pups received 15 mg/kg of AZD4831 at P2, P4, P6, P8, and P10. The irreversible MPO inhibitor does not affect the morphology of rat lungs raised in room air or the BPD lungs. Scale bar = 100 μm. *: *p* < 0.054 (n = 9).

**Table 2 antioxidants-13-00889-t002:** Literature reported risk factors for BPD.

Prenatal	At Birth	Postnatal
Intrauterine growth restriction [45,46]	Gestational age and birth weight [1,47]	Respiratory support [48,49,50,51,52]
Maternal smoking [53]	Gender [54,55]	Infection [56,57]
Lack of antenatal corticosteroid [58,59]	Level of management [60]	Patent ductus arteriosus [61,62,63,64,65]
Chorioamnionitis [66,67,68]		Gastroesophageal reflux [69,70]
Genetics [71,72,73,74]		

## Data Availability

The original contributions presented in the study are included in the article, further inquiries can be directed to the corresponding author.

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
