# Peer review of "Role of Myeloperoxidase, Oxidative Stress, and Inflammation in Bronchopulmonary Dysplasia"

_antioxidants, 2024, doi:10.3390/antiox13080889_

Round 1
Reviewer 1 Report
The review has great depth and is overall an excellent summary of the current knowledge surrounding the leading causes of Bronchopulmonary dyspepsia with a fine focus on the current and previous work of the authors. The manuscript is ready for publication in its current state following some minor edits by the editorial team.
Statements at line 77 and line 100 may require revisiting, to more accurately portray the ideas wished to highlight.
Author Response
Thanks for your kind comments. I asked all team members to edit for grammar, and we used Grammarly as a final editing check. We declared that we used this program after finalizing the draft before our first submission. In addition, we look to the editorial team, per the reviewers' suggestion, for a final check.
Comments 1: Statements at line 77 and line 100 may require revisiting, to more accurately portray the ideas wished to highlight.
Response 1: We have revisited the statements as you kindly suggested. We have explained why premature births keep increasing (lines 83-87) and added the improved survival rate in extremely premature neonates as one reason why BPD prevalence remains unchanged (lines 108-111).
Reviewer 2 Report
Tzong-Jin Wu et al. submitted an interesting review about BPD. The topic was of a certain significance, and would arouse a certain impact in its field, if accepted. The paper fell within the scope of Antioxidants. Overall, the manuscript quality was good, and could be considered for publication after a Minor Revision.
Tzong-Jin Wu et al. submitted an interesting review about BPD. The topic was of a certain significance, and would arouse a certain impact in its field, if accepted. The paper fell within the scope of Antioxidants. Overall, the manuscript quality was good, and could be considered for publication after a Minor Revision. Detailed comments:
1. How much did BPD management cost globally? This could be added at the beginning of Introduction Section.
2. The logical structure of the review should be outlined at the end of Introduction Section.
3. Apart from criteria in Table 1, were there any criteria from established official standards for BPD diagnosis? Please discuss.
4. For literature reported the risk factors for BPD (Table 2), a brief bibliometric analysis could be performed.
5. In Section 8, some novel nanotechnology-based therapy could also be discussed for BPD. For instance, liposome-based therapeutics.
6. Some social perspectives for BPD treatment could be briefly supplemented in Discussion Section, such as the government monitoring and administration, hospital vigilance system, pharmacoeconomics and sustainable development goals (SDGs).
7. The Conclusion Section was a bit short. Please consider to add some personal opinions for future directions. For example, what could the science community do to improve the therapy of BPD?
8. Ref#148: It might be improper to cite a website in this manner.
Author Response
Thanks for your kind words.
Comment 1: How much did BPD management cost globally? This could be added at the beginning of Introduction Section.
Response 1: We couldn’t find a report about the global cost of BPD management. Humayun et al.'s systemic review is added to the introduction. We hope their findings can help readers realize the financial impact of BPD (lines 37-42).
Comment 2: The logical structure of the review should be outlined at the end of Introduction Section.
Response 2: We agree with your suggestion and revised the introduction to include the logical structure of the review at the end (lines 120-130).
Comment 3: Apart from criteria in Table 1, were there any criteria from established official standards for BPD diagnosis? Please discuss.
Response 3: We probably did not include all diagnostic criteria from the literature, but we believe the most commonly used ones are detailed. We briefly discussed the issues of implementing different versions of diagnostic criteria. However, the section intends to show the evolution over time when researchers attempt to modify them for various reasons.
Comment 4: For literature reported the risk factors for BPD (Table 2), a brief bibliometric analysis could be performed.
Response 4: We have added the information to the table as suggested.
Comment 5: In Section 8, some novel nanotechnology-based therapy could also be discussed for BPD. For instance, liposome-based therapeutics.
Response 5: This review centers on MPO-induced oxidative stress and sterile inflammation in the BPD destructive cycle, so we did not include other antioxidant or anti-inflammatory therapies. We agree that several novel therapeutic strategies are worth mentioning, but they are outside the scope of this review. Your suggestion will be considered in future review articles about antioxidant and anti-inflammatory treatments for BPD.
Comment 6: Some social perspectives for BPD treatment could be briefly supplemented in Discussion Section, such as the government monitoring and administration, hospital vigilance system, pharmacoeconomics and sustainable development goals (SDGs).
Response 6: This suggestion is beyond our original intent. However, we have to agree that it is essential to inform the readers about the difficulties we face during drug development for neonates. The revised manuscript has added a new paragraph (lines 681-693).
Comment 7: The Conclusion Section was a bit short. Please consider to add some personal opinions for future directions. For example, what could the science community do to improve the therapy of BPD?
Response 7: Thank you for pointing it out. Reviewer #3 provided the same suggestion. We have added our opinions and limitations to the conclusion.
Comment 8: Ref#148: It might be improper to cite a website in this manner.
Response 8: We apologize for incorrectly citing this paper. Reference 148 has been changed to correctly reflect the authors, title, and journal. We thank the reviewer for drawing our attention to this error.
Reviewer 3 Report
In this review article entitled “Role of Myeloperoxidase, Oxidative Stress, and Inflammation in Bronchopulmonary Dysplasia”, the authors summarized the role of oxidative stress in the development of bronchopulmonary dysplasia (BPD), as well as the potential mechanisms by which anti-myeloperoxidase (MPO) therapy attenuates BPD. Overall, the manuscript is comprehensive and provides a deep insight into the BPD. It is well-written and easy to read. Several key sections are well illustrated by figures. The references are properly cited in the manuscript. The major concern is that original data were presented in this review article, which is not allowed according to the journal’s guidance.
Detailed comments are listed as follows:
1. Some sections appear not close to the topic, it would be better to remove them to focus on the pathogenesis and mechanisms. For example, Biomarkers for BPD.
2. The font is not consistent. For example, lines 43-44 are Arial.
3. Research data should be removed given it’s a review article in nature.
4. It would be better to include the current limitations and future directions at the end of the Conclusion section.
Author Response
Thank you so much for your positive feedback. We understand it is policy not to include original data in the review article. However, we strongly felt that the story is/was incomplete without the new information. For this reason, we consulted the editorial office before accepting the invitation, and sought and received permission to include new data in this article for the reasons already cited.
Comment 1: Some sections appear not close to the topic, it would be better to remove them to focus on the pathogenesis and mechanisms. For example, Biomarkers for BPD.
Response 1: We understand and appreciate the reviewer’s perspective. However, part of our intent in the review was to convey the progress being made in BPD but also to highlight the considerable limtations of our understanding. We believe that the biomarker section and the paucity of viable biomarkers in all aspects of BPD strongly highlights our point. In addition the editorial office approved the outlines of this review, including biomarkers for BPD, before we started to prepare it. We consider the information about biomarkers for BPD interesting to some readers as research into biomarkers has become a hot topic in several governmental funding agencies and is badly needed in BPD.
Comment 2: The font is not consistent. For example, lines 43-44 are Arial.
Response 2: The font has been changed to Palatino Linotype for lines 48-49.
Comment 3: Research data should be removed given it’s a review article in nature.
Response 3: The editorial office approved the addition of the unpublished data so that we could explain how myeloperoxidase activity contributes to BPD. The approval was discussed before we accepted the invitation. We firmly believe the KYC mechanism will not be complete without adding the unpublished data.
Comment 4: It would be better to include the current limitations and future directions at the end of the Conclusion section.
Response 4: As you suggested, we have added them to the conclusion section (lines 775-783). We hope this change adequately addresses your suggestion.
Round 2
Reviewer 3 Report
All my concerns have been addressed properly.
No additional comments on the manuscript.